# Estimated prevalence and gender disparity of physical activity among 64,127 in-school adolescents (aged 12–17 years): A multi-country analysis of Global School-based Health Surveys from 23 African countries

**Martin Ackah**[1]*, **David Owiredu**[2], **Mohammed Gazali Salifu**[3], **Cynthia Osei Yeboah**[1]

**1** Department of Physiotherapy, Korle Bu Teaching Hospital, Accra, Ghana, **2** Centre for Evidence synthesis, University of Ghana, Accra, Ghana, **3** Policy Planning Budgeting Monitoring and Evaluation Directorate, Ministry of Health, Accra, Ghana

* martinackah10@gmail.com

**Data Availability Statement:** This study conducted a secondary analysis of third party data collected

## Abstract

The Africa sub-region currently lacks quantitative normative data to illustrate the extent of burden and gender inequities of physical activity level in order to inform policy and education, towards meeting the WHO's 2030 physical activity milestone. The study aimed to provide insights on the current prevalence of sufficient physical activity and gender disparity, using a nationally representative data from the Global School-based student Health Survey (GSHS) from 23 African countries. The study used the multi-country GSHS data from 23 African countries (2003–2017). Sufficient physical activity was measured through self-administered questionnaire. The prevalence of sufficient physical activity among in-school adolescents in each country was estimated by proportion with corresponding 95% confidence intervals. Meta-analysis with random effect was employed to pool the prevalence of physical activity level in the 23 African countries. Additionally, sub-group, sensitivity, and meta-regression analyses were performed. The study included 23 African countries representing 64,127 in-school adolescents aged 12–17 years. Overall, only 20% [95% CI: 18%-22%] of adolescents in Africa engaged in sufficient physical activity. With respect to sex, only 25% [95% CI: 22%-28%] of males and 16% [95% CI: 14%-18%] of females met the WHO recommendation of sufficient physical activity. Sufficient physical activity ranged from 11.6% [9.2%-14.5%] in Sudan to 38.3% [CI:30.2%-47.1%] in Benin. Sufficient physical activity in boys ranged from 7.5% [95% CI: 6.2%-9.0%] in Zambia to 29.2% [95% CI: 22.5%-36.8%] in Benin, and ranged from 2.5% [95% CI: 1.6%-4.0%] in Senegal to 12.2% [95% CI:10.6%-14.1%] in Tanzania for girls. Only 20% of in-school adolescents met the WHO's recommended physical activity level. Generally, adolescent girls in Africa are less active than adolescent boys. Addressing the rising burden of insufficient physical activity in adolescents and narrowing the gender gap could ultimately increase the overall physical activity engagement and achieve the WHO's global physical activity target by 2030.

from WHO's publicly available Global School-based Students Health Surveys (https://extranet.who.int/ncdsmicrodata/index.php/catalog/centraltools/). The authors confirm that they did not have any special access or request privileges that others would not have.

**Funding:** The authors received no specific funding for this work.

**Competing interests:** The authors have declared that no competing interests exist.

## Introduction

Physical activity is described as any skeletal muscle movement that involves energy expenditure, such as walking, cycling, sports, and other active leisure activities [1]. Despite enormous benefits on health (i.e., improve cardiovascular fitness, muscle strength, bone health, and reduce depression), cognitive development, and prosocial behavior in people aged 11–17 years, the majority of adolescents are considered to be physically inactive globally [2,3]. The World Health Organization (WHO) estimates that over five million deaths could be prevented each year if the world's population was physically active [4]. In light of this, the WHO emphasized the importance of physical activity at the 2018 World Health Assembly, setting a goal to reduce insufficient physical activity among adolescents by 15% by 2030 [1,2].

Global and regional observational studies have estimated and delved into the descriptive epidemiology of physical activity among adolescents. For instance, Hallal et al estimated that a high proportion (i.e., 80.3%) of 13–15-year-olds never met the WHO physical activity recommendation in 2012 [5]. Aguilar-Farias and colleagues found that only 15% of Latin American and Caribbean adolescents were physically active (engaged in at least 60 minutes of moderate-to-vigorous physical activity each day) [6]. Similarly, a study conducted by Xu and colleagues in 54 low- and middle-income countries found that the prevalence of physical activity among adolescents aged 12 to 15 years is 15.2% [7] Furthermore, McMahon and co-authors showed that 13.6 percent of their study's sample was adequately active in a ten-country survey of European adolescents [8]. Additionally, Guthold and co-authors estimated that more than 80% of school-children aged 11–17 years worldwide never met the WHO-recommended level of physical activity in 2016 [2]. In Africa, pocket of studies has been conducted at the country-level. Seidu et al., for example, reported that a large proportion of Ghanaian in-school adolescents were physically inactive [9]. The fundamental challenge was the studies' ability to extrapolate findings to the current African context. For example, Guthold and colleagues' global estimates study did not include Liberia, Mauritius and Sierra Leone GSHS data. Additionally, a stratified pooled analysis was not performed for Africa, therefore a representative prevalence estimates for physical activity among adolescents aged 12 to 17 years on the continent is currently unknown.

In a recent systematic review of studies conducted in sub-Saharan Africa, urbanization was substantially associated with declining and increasing trends in physical activity and sedentary behavior [10]. Additionally, the authors recommended that future works should include coordinated efforts to conduct nationally representative surveys using comparable or common measurement techniques, sampling procedures, and multi-country surveys in order to effectively monitor physical activity transitions over time in this region [10]. The GSHS is a multi-country survey among adolescents that identifies young people's health behavior and protective variables (i.e., physical activity level).

Physical activity is recognized to be influenced by socio-demographic characteristics such as sex, age, and socioeconomic status [11]. However, Ricardo et al mentioned that sex as a variable was a major contributor to the disparities in physical activity, in that boys were more active than girls [12]. Therefore, it is anticipated that a 4.8 percentage point reduction in global levels of inactivity among females would be adequate to meet the WHO's target of a 15% reduction in global levels of inactivity by 2030 [1,2]. The Africa sub-region currently lacks quantitative normative data to illustrate the extent of gender inequalities in order to inform policy and education, towards meeting the WHO's 2030 physical activity milestone.

The lack of representative data on adolescents' (12 to 17 years) physical activity level in Africa is likely to impede the designing of effective public health policy, working interventions and decision making, as well as undermine efforts to achieve the WHO's target for 15%

reduction in physical inactivity by 2030 [1,2] on the continent. Hence, the current study aimed to provide the overall baseline data and further explore the gender disparity on sufficient physical activity level among in-school adolescents in Africa using a nationally representative data from the GSHS from 23 African countries.

## Methods

### Study design, sampling, and data source

The current study strictly followed Strengthening the Reporting of Observational Studies in Observational Studies in Epidemiology (STROBE) guidelines [13]. The data for this study were sourced from the GSHS, which was conducted across 23 African countries among in-school adolescents from 2003–2017. The survey gathered information on health behaviors and risk factors related to the world's top cause of death and mortality among children and adolescents. The GSHS is a joint effort between WHO, the Centers for Disease Control and Prevention in the United States, and participating countries. Detailed methodology can be found at https://www.who.int/teams/noncommunicable-diseases/surveillance/systems-tools/global-school-based-student-health-survey/methodology.

The GSHS used a 2-stage cluster sampling technique. For example, schools were chosen in the first stage based on a probability proportional to the number of pupils enrolled. After then, a random class was chosen, and every student in that class was eligible to participate in the study.

The GSHS received ethical approval from each country's Ministry of Health or Education as well as an institutional review board or ethics committee. In all circumstances, a consent/assent form was obtained from the children and/or parents, as well as the schools, depending on the country laws and stipulations.

### Measure

Physical activity was assessed by a single question and was self-reported. The question was *"During the past 7 days, on how many days were you physically active for a total of at least 60 minutes per day?"* The responses ranged from zero day to seven days (0–7 days). Based on WHO recommendations and evidence from previous studies [4,6,9,14,15], an adolescent was considered as 'physically active' if he/she engages in physical activity for at least 60 minutes per day for at least five days in a week. As a result, the outcome was dichotomized as; "sufficient physical activity" [i.e., met the recommended ≥5 days per week of physical activity] and "insufficient physical activity" [i.e., did not meet the recommended ≥5 days per week of physical activity] [9].

### Statistical analysis

To reflect the weighting method and the two-stage sampling design, we appended weights, strata, and a primary sampling unit (PSU) to every student record in the GSHS data file. The weights allow the results to be extrapolated to the entire study population as well as the entire in-school adolescents' population in Africa. The stratum represents the first stage (school level) of the GSHS sample, whereas the PSU represents the second step (classroom level). STATA was used for all of the analyses (Stata Statistical Software: Release 16; College Station, TX; Stata Corp LP).

The prevalence of sufficient physical activity among in-school adolescents in each country was estimated by proportion with corresponding 95% Confidence Interval (CI). Frequencies and percentages were used to describe the data. Meta-analysis with random effect was

employed to pool the overall prevalence of sufficient physical activity level among in-school adolescent in Africa. Additionally, gender-specific estimates were generated for each country in Africa separately through random effect meta-analysis. Both the overall and gender-specific estimates of sufficient physical activity were represented on forest plots. The essence of the meta-analysis was to cater for any heterogeneity that might have arisen from the data from the various countries. Heterogeneity was assessed using the I-squared statistic [16].

A sub-group analysis was conducted on the Human Development Index (low vs medium vs high vs very high), economic health (low-income countries vs. lower middle-income countries vs. upper middle-income countries vs high income countries), sub-region (North Africa vs West Africa vs Southern Africa vs East Africa), and survey year (2003–2010 vs 2011–2017). Leave one out sensitivity analysis was conducted to assess the robustness of the estimates obtained from the meta-analysis. Finally, a multivariable meta-regression analysis was performed to determine the sources of heterogeneity in the current study.

## Results

### Background characteristics

The study included 23 African countries representing 64,127 in-school adolescents aged 12–17 years. The survey years ranged from 2003 in Uganda and Zimbabwe to 2017 in Liberia, Mauritius, and Sierra Leone. Sample size ranged from 1,951 in Mauritania to 6,345 in Morocco. The response rate ranged from 60% in Senegal to 98% in Algeria and Libya. The male adolescent participants were highest in Senegal (55.1%) and lowest in Sudan (38.8%). The results are displayed in Table 1.

### Prevalence of physical activity among adolescents in Africa

Overall, only 20% [95% CI: 18%-22%] of adolescents in Africa engaged in physical activity (Fig 1). With respect to sex, only 25% [95% CI: 22%-28%] of the male adolescents met the WHO recommendation of sufficient physical activity (Fig 2) whilst 16% [95% CI: 14%-18%] of the female adolescents met the WHO recommendation of sufficient physical activity (Fig 3).

Across Africa, the majority of the in-school adolescents did not engage in sufficient physical activity. For example, sufficient physical activity ranged from 11.6% [9.2%-14.5%] in Sudan to 38.3% [CI:30.2%-47.1%] in Benin.

With regards to sex, sufficient physical activity in boys ranged from 7.5% [95% CI: 6.2%-9.0%] in Zambia to 29.2% [95% CI: 22.5%-36.8%] in Benin, and ranged from 2.5% [95% CI: 1.6%-4.0%] in Senegal to 12.2% [95% CI:10.6%-14.1%] in Tanzania for girls (Table 2). The country with large absolute sex gap was Benin (20.1 percentage points).

Nevertheless, some countries such as Botswana (8.9% vs 7.7%), Ghana (10.9% vs 8.8%), Namibia (11.1% vs 10.9%), Sudan (6.2% vs 5.3%), Uganda (10.9% vs 8.9%), Zambia (7.5% vs 7.5%), and Zimbabwe (9.1% vs 8.6%) have similar or narrowed gender gap with respect to sufficient physical activity engagement.

### Sub-group and sensitivity analyses

The sub-group analysis as shown Fig 4 was based on HDI, economic health, sub-region and the survey years. With respect to HDI, the pooled estimates of sufficient physical activity from countries with low, medium, high, and very high HDI were; 20.0% [95% CI: 15.0%-24,0%], 18.0% [95% CI: 16.0%-21,0%], 20.0% [95% CI: 18.0%-22,0%], and 30.0% [95% CI: 26.0%-33.0%] respectively. With regards to economic health, the pooled estimates of sufficient physical activity from countries with low income, lower-middle income, upper middle income, and

**Table 1. Characteristics of the adolescents included in the GSHS from 23 African countries (n = 64,127).**

| SN | Country | Survey Year | Response rate (%) | Adolescents meeting recommended WHO PA level | Sample size | Male (%) |
|----|---------|-------------|-------------------|----------------------------------------------|-------------|----------|
| 1 | Algeria | 2011 | 98 | 906 | 4325 | 48.3 |
| 2 | Benin | 2016 | 78 | 552 | 1562 | 48.5 |
| 3 | Botswana | 2005 | 95 | 353 | 2174 | 55.0 |
| 4 | Djibouti | 2007 | 83 | 353 | 1733 | 57.0 |
| 5 | Egypt | 2012 | 85 | 407 | 2413 | 46.7 |
| 6 | Ghana | 2012 | 71 | 743 | 3517 | 53.8 |
| 7 | Kenya | 2003 | 89.4 | 682 | 3398 | 49.0 |
| 8 | Liberia | 2017 | 71 | 436 | 2330 | 52.7 |
| 9 | Libya | 2007 | 98 | 398 | 2060 | 43.6 |
| 10 | Mauritania | 2010 | 70 | 383 | 1951 | 47.3 |
| 11 | Mauritius | 2017 | 84 | 879 | 2945 | 47.2 |
| 12 | Morocco | 2016 | 91 | 942 | 6345 | 53.2 |
| 13 | Mozambique | 2015 | 80 | 356 | 1752 | 53.4 |
| 14 | Namibia | 2013 | 89 | 951 | 4401 | 47.3 |
| 15 | Senegal | 2005 | 60 | 359 | 2915 | 55.1 |
| 16 | Seychelles | 2015 | 82 | 580 | 2402 | 47.5 |
| 17 | Sierra Leone | 2017 | 82 | 671 | 2667 | 45.8 |
| 18 | Sudan | 2012 | 77 | 262 | 2137 | 38.8 |
| 19 | Tanzania | 2014 | 87 | 439 | 3503 | 48.6 |
| 20 | Tunisia | 2008 | 83 | 613 | 2775 | 48.6 |
| 21 | Uganda | 2003 | 69 | 627 | 3072 | 51.4 |
| 22 | Zambia | 2004 | 70 | 266 | 1824 | 49.1 |
| 23 | Zimbabwe | 2003 | 84 | 349 | 1926 | 44.0 |

PA = Physical Activity.

high- income economic health were; 19.0% [95% CI: 15.0%-24,0%], 19.0% [95% CI: 16.0%-22,0%], 22.0% [95% CI: 16.0%-28,0%], and 24.0% [95% CI: 20.0%-28,0%]. Again, within the sub-region, the pooled sufficient physical estimates for North Africa was18.0% [95% CI: 15.0%-21,0%], for 22.0% [95% CI: 16.0%-28.0%] for West Africa, 19.0% [95% CI: 14.0%-24.0%] for Southern Africa was 19.0% [95% CI: 14.0%-24.0%], and 20.0% [95% CI: 17.0%-24.0%] for Eastern Africa was 20.0% [95% CI: 17.0%-24.0%]. Finally, for GSHS survey year, the pooled estimates for survey conducted from 2003–2010 and 2011–2017 were; 18.0% [95% CI: 16.0%-20.0%], and 21.0% [95% CI: 17.0%-24,0%] respectively. Additionally, the authors performed sensitivity analyses by removing one survey at a time and recalculating the pooled effect estimate of the remaining surveys on the pooled burden of sufficient physical activity. The results showed that there is no influential survey on the overall pooled estimate. For instance, the pooled estimated sufficient physical activity level among adolescents in African ranged from 19.0% [95% CI: 17.0%-21.0%] in Benin and Mauritius to 20.0% [95% CI: 17.0%-22.0%] (S1 Table).

## Meta-regression analysis

To explore the sources of heterogeneity that existed in estimated physical activity prevalence, meta-regression analysis was performed. The findings showed significant increased trend in the GSHS survey years (coefficient = 0.005, p = 0.022). However, insignificant trends were found for HDI, economic health, sub-region, and the sample size [p>0.05] (S2 Table).

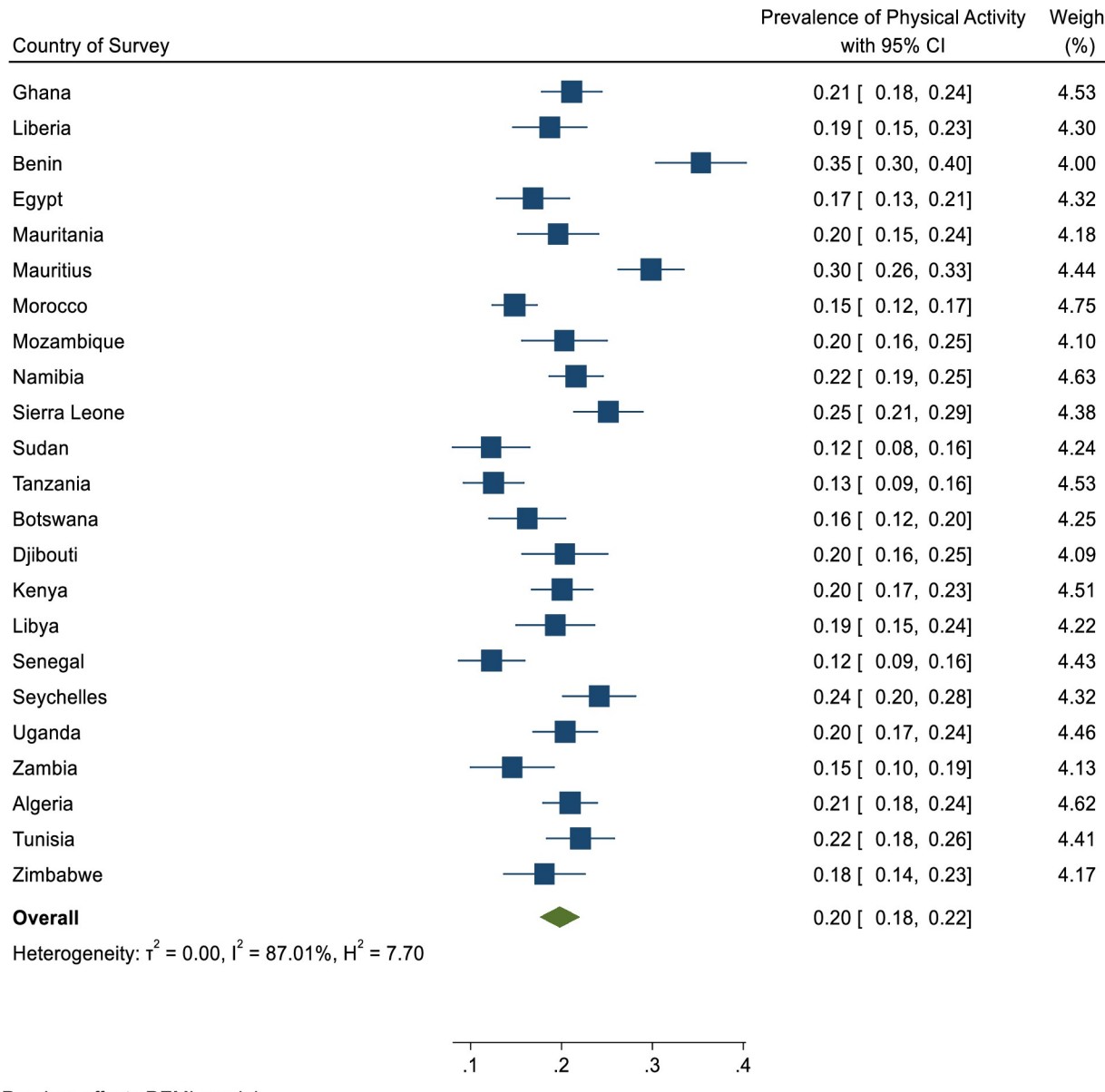

| Country of Survey | Prevalence of Physical Activity with 95% CI | Weight (%) |
|---|---|---|
| Ghana | 0.21 [ 0.18, 0.24] | 4.53 |
| Liberia | 0.19 [ 0.15, 0.23] | 4.30 |
| Benin | 0.35 [ 0.30, 0.40] | 4.00 |
| Egypt | 0.17 [ 0.13, 0.21] | 4.32 |
| Mauritania | 0.20 [ 0.15, 0.24] | 4.18 |
| Mauritius | 0.30 [ 0.26, 0.33] | 4.44 |
| Morocco | 0.15 [ 0.12, 0.17] | 4.75 |
| Mozambique | 0.20 [ 0.16, 0.25] | 4.10 |
| Namibia | 0.22 [ 0.19, 0.25] | 4.63 |
| Sierra Leone | 0.25 [ 0.21, 0.29] | 4.38 |
| Sudan | 0.12 [ 0.08, 0.16] | 4.24 |
| Tanzania | 0.13 [ 0.09, 0.16] | 4.53 |
| Botswana | 0.16 [ 0.12, 0.20] | 4.25 |
| Djibouti | 0.20 [ 0.16, 0.25] | 4.09 |
| Kenya | 0.20 [ 0.17, 0.23] | 4.51 |
| Libya | 0.19 [ 0.15, 0.24] | 4.22 |
| Senegal | 0.12 [ 0.09, 0.16] | 4.43 |
| Seychelles | 0.24 [ 0.20, 0.28] | 4.32 |
| Uganda | 0.20 [ 0.17, 0.24] | 4.46 |
| Zambia | 0.15 [ 0.10, 0.19] | 4.13 |
| Algeria | 0.21 [ 0.18, 0.24] | 4.62 |
| Tunisia | 0.22 [ 0.18, 0.26] | 4.41 |
| Zimbabwe | 0.18 [ 0.14, 0.23] | 4.17 |
| **Overall** | 0.20 [ 0.18, 0.22] | |

Heterogeneity: $\tau^2 = 0.00$, $I^2 = 87.01\%$, $H^2 = 7.70$

Random-effects REML model

**Fig 1. Pooled estimated prevalence of sufficient physical activity level across 23 African countries.**

## Discussion

The current study estimated the prevalence and gender disparity of physical activity among in-school adolescents aged 12–17 years using multi-country analysis of GSHS data from 23 African countries. The study was designed to provide the overall baseline data and further explore the gender disparity on sufficient physical activity level among in-school adolescents in Africa.

Overall, 20% of adolescents in Africa met the current WHO physical activity recommendation. This finding, for example, is in line with the findings of earlier global investigations [2,5]. Physical inactivity among adolescents aged 11 to 17 years was shown to be significantly high in the global pooled study of 298 population-based surveys comprising 1.6 million adolescents [2]. Xu et al and McMahon et al also reported similarly high rates of physical inactivity in

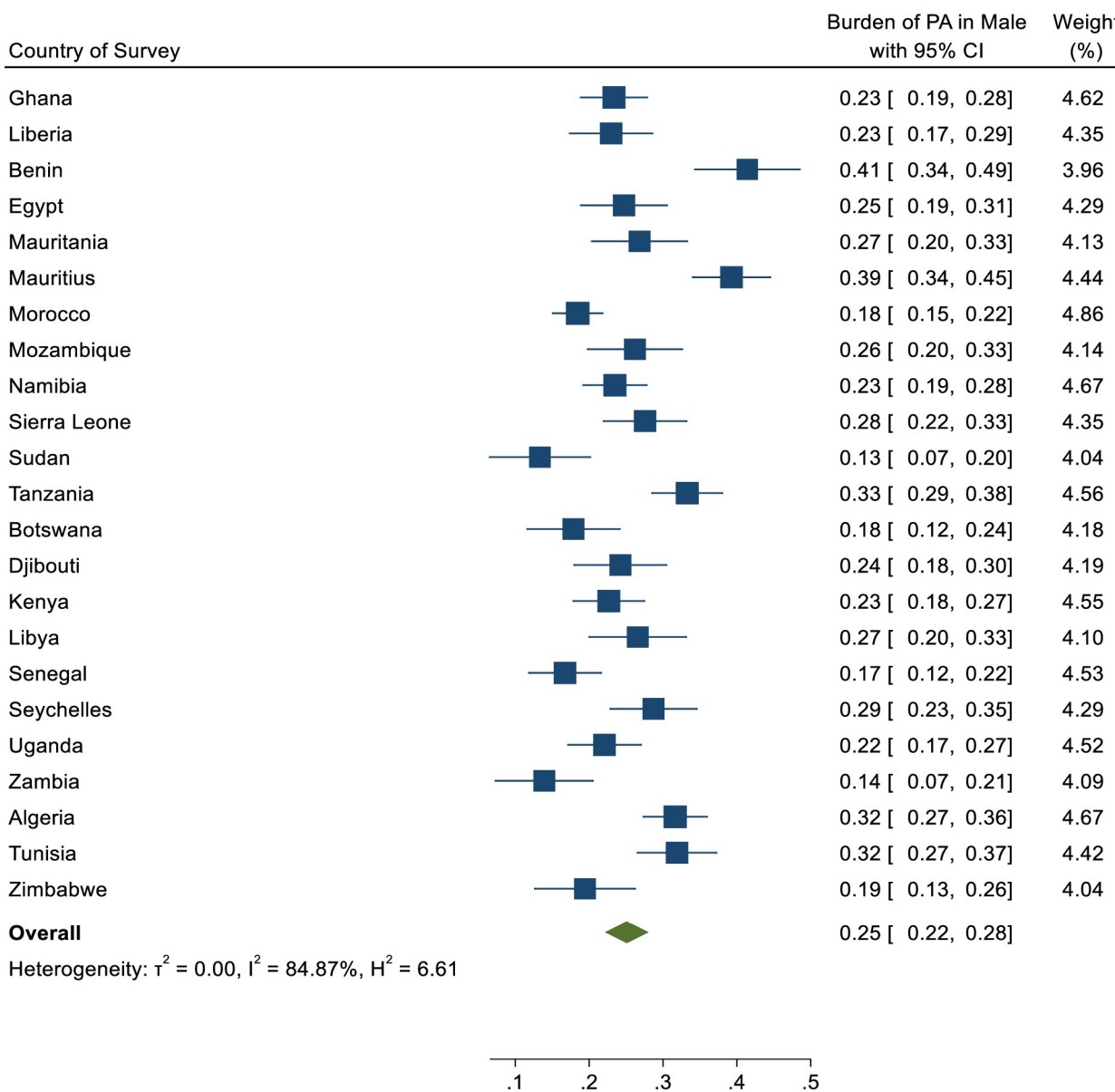

| Country of Survey | | Burden of PA in Male with 95% CI | Weight (%) |
|---|---|---|---|
| Ghana | | 0.23 [ 0.19, 0.28] | 4.62 |
| Liberia | | 0.23 [ 0.17, 0.29] | 4.35 |
| Benin | | 0.41 [ 0.34, 0.49] | 3.96 |
| Egypt | | 0.25 [ 0.19, 0.31] | 4.29 |
| Mauritania | | 0.27 [ 0.20, 0.33] | 4.13 |
| Mauritius | | 0.39 [ 0.34, 0.45] | 4.44 |
| Morocco | | 0.18 [ 0.15, 0.22] | 4.86 |
| Mozambique | | 0.26 [ 0.20, 0.33] | 4.14 |
| Namibia | | 0.23 [ 0.19, 0.28] | 4.67 |
| Sierra Leone | | 0.28 [ 0.22, 0.33] | 4.35 |
| Sudan | | 0.13 [ 0.07, 0.20] | 4.04 |
| Tanzania | | 0.33 [ 0.29, 0.38] | 4.56 |
| Botswana | | 0.18 [ 0.12, 0.24] | 4.18 |
| Djibouti | | 0.24 [ 0.18, 0.30] | 4.19 |
| Kenya | | 0.23 [ 0.18, 0.27] | 4.55 |
| Libya | | 0.27 [ 0.20, 0.33] | 4.10 |
| Senegal | | 0.17 [ 0.12, 0.22] | 4.53 |
| Seychelles | | 0.29 [ 0.23, 0.35] | 4.29 |
| Uganda | | 0.22 [ 0.17, 0.27] | 4.52 |
| Zambia | | 0.14 [ 0.07, 0.21] | 4.09 |
| Algeria | | 0.32 [ 0.27, 0.36] | 4.67 |
| Tunisia | | 0.32 [ 0.27, 0.37] | 4.42 |
| Zimbabwe | | 0.19 [ 0.13, 0.26] | 4.04 |
| **Overall** | | 0.25 [ 0.22, 0.28] | |

Heterogeneity: $\tau^2 = 0.00$, $I^2 = 84.87\%$, $H^2 = 6.61$

Random-effects REML model

**Fig 2. Pooled estimated prevalence of sufficient physical activity level across 23 African countries stratified by male gender.** Burden as used in Fig 2 is a general term for prevalence in burden of disease studies or epidemiological studies.

adolescents in their respective multinational studies [7,8]. These estimates present a worrying prospect given the implications of sedentary behavior and physical inactivity on the development of coronary heart diseases and a host of other non-communicable diseases [17].

The pooled prevalence of physical activity level was higher among boys than girls in Africa 25% [95% CI: 22%-28%] vs. 16% [95% CI: 14%-18%] respectively. With regards to country-specific prevalence of physical activity, 22/23 countries have boys' physical activity level exceeding that of the adolescent girls. This finding corroborates the findings of several previous studies. For example, the Lancet physical activity research working group presented data on children and adolescents in 105 countries in 2012, showing that insufficient physical

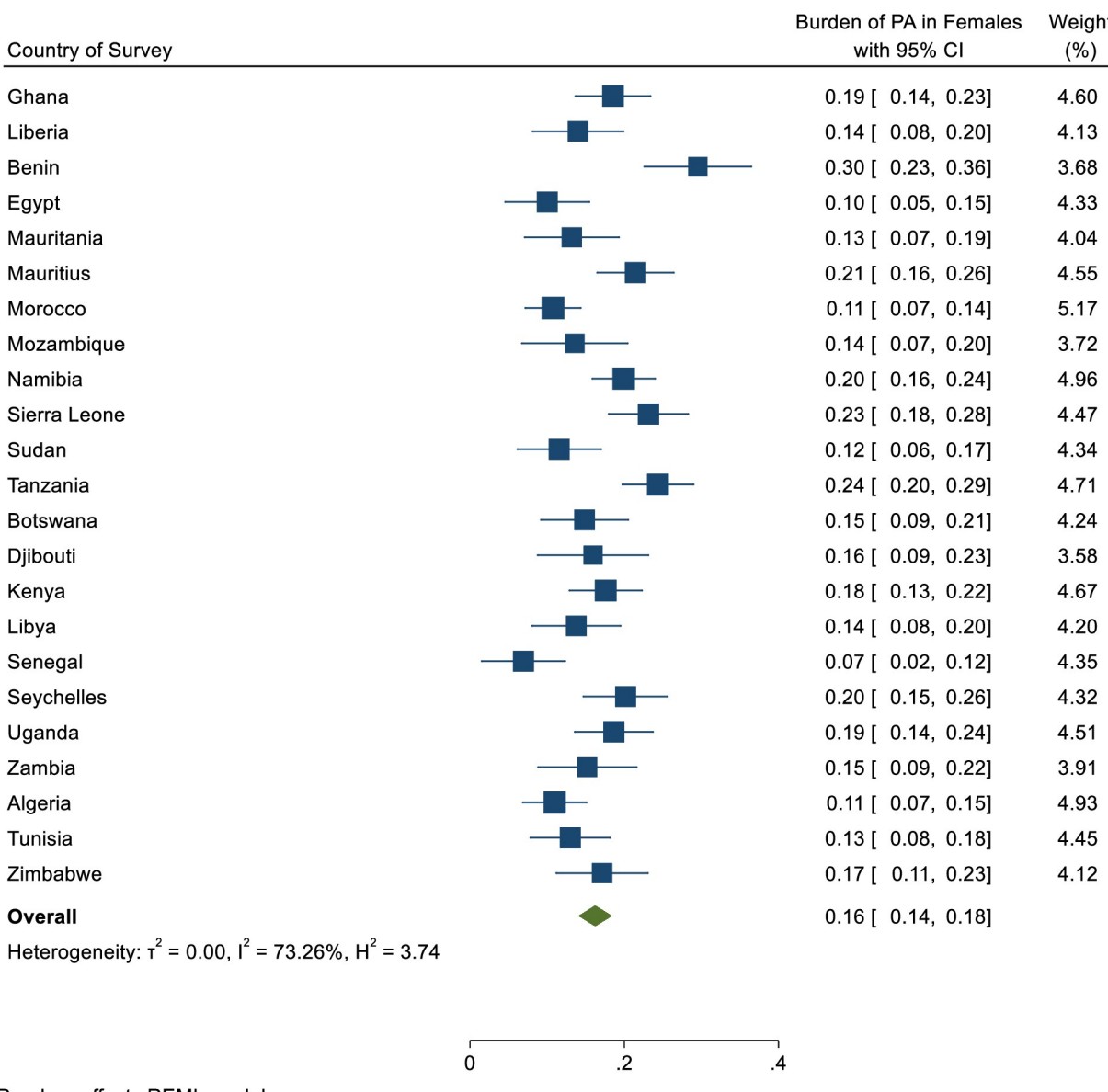

**Fig 3. Pooled estimated prevalence of sufficient physical activity level across 23 African countries stratified by female gender.** Burden as used in Fig 3 is a general term for prevalence in burden of disease studies or epidemiological studies.

activity worsened with age, particularly among girls [5]. Similarly, Guthold and colleagues asserted that the frequency of female insufficient physical activity has not altered globally, across socioeconomic regions, and in practically all of the countries analyzed since 2001 [2].

In our sample, widening gender gap were particularly pronounced in Benin, Algeria, Egypt, Libya Tunisia, and Mozambique. Generally, Ricardo and colleagues attributed this widening gap to stereotypes, body image insecurity, and cultural acceptability due to intrinsic sociocultural norms hence making girls enjoy less and have the confidence to engage in sport related activities [12]. Additionally, it was observed that most north African countries largely form the widening gender gap with respect to physical activity participation. This could be ascribed to various traditional and religious gender norms that persist in these countries such as the need

**Table 2. Country-specific prevalence of physical activity among adolescents in 23 African countries.**

| Country | Prevalence of PA in boys [95% CI] | Prevalence of PA in girls [95% CI] | Overall Prevalence of PA [95% CI] |
|---|---|---|---|
| Algeria | 15.2 [13.8–16.9] | 5.9 [4.9–6.9] | 21.1 [19.5–22.8] |
| Benin | 29.2 [22.5–36.8] | 9.1 [6.9–11.93] | 38.3 [30.2–47.1] |
| Botswana | 8.9 [7.4–10.8] | 7.7 [6.8–8.7] | 16.7 [15.3–18.3] |
| Djibouti | 14.8 [13.5–16.3] | 6.4 [5.2–7.8] | 21.2 [19.7–22.9] |
| Egypt | 11.9 [8.5–16.5] | 5.0 [3.5–7.2] | 16.9 [13.2–21.5] |
| Ghana | 10.9 [9.4–12.7] | 8.8 [6.9–11.0] | 19.7 [16.8-22-9] |
| Kenya | 11.1 [9.2–13.5] | 9.2 [7.5–11.3] | 20.4 [17.8–23.2] |
| Liberia | 11.6 [9.4–14.2] | 6.9 [5.7–8.6] | 18.6 [15.4–22.3] |
| Libya | 13.7 [11.4–16.5] | 6.8 [5.2–8.9] | 20.6 [18.2–23.1] |
| Mauritania | 14.5 [11.2–18.6] | 5.9 [4.6–7.5] | 20.5 [16.3–25.5] |
| Mauritius | 17.9 [11.8–26.4] | 11.5 [8.0–16.1] | 29.5 [256–33.6] |
| Morocco | 10.4 [9.4–11.5] | 5.0 [4.5–5.6] | 15.5 [14.3–16.7] |
| Mozambique | 13.8 [10.5–18.1] | 6.5 [4.4–9.6] | 20.4 [15.4–26.4] |
| Namibia | 11.1 [9.8–12.5] | 10.9 [9.7–12.2] | 22.0 [19.9–24.2] |
| Senegal | 9.9 [7.1–13.6] | 2.5 [1.6–4.0] | 12.5 [9.0–16.9] |
| Seychelles | 14.0 [12.2–16.1] | 10.4 [8.9–12.7] | 24.5 [22.3–26.7] |
| Sierra Leone | 15.2 [12.9–17.7] | 11.9 [8.1–17.3] | 27.2 [22.9–31.8] |
| Sudan | 6.2 [3.8–10.2] | 5.3 [3.2–8.7] | 11.6 [9.2–14.5] |
| Tanzania | 15.9 [13.3–19.1] | 12.2 [10.6–14.1] | 28.2 [25.0–31.7] |
| Tunisia | 16.1 [14.3–17.9] | 6.5 [5.3–7.9] | 22.5 [19.8–25.5] |
| Uganda | 10.9 [8.3–14.3] | 8.9 [7.1–11.3] | 19.9 [17.4–22.7] |
| Zambia | 7.5 [6.2–9.0] | 7.5 [5.9–9.5] | 14.9 [13.2–16.9] |
| Zimbabwe | 9.1 [7.7–10.7] | 8.6 [5.5–13.4] | 17.7 [14.3–21.8] |

PA = Physical Activity, CI = Confidence Interval.

for unmarried women to be accompanied in public places, conservative attires unsuitable for related physical activity engagement, and scarcity and accessibility of specific gender fitness facilities due to safety concerns [18]. Consequently, there are greater freedom for boys to engage more in physical activity than their girl counterparts [19]. Addressing the rising burden of insufficient physical activity physical activity in adolescent girls and narrowing the gender gap in Africa could commence with offering more opportunities for safe and, better access to physical activity-based interventions, and through advocacy investment and for shifting socio-cultural norms. This will ultimately increase the overall physical activity engagement and drive the achievement of WHO's global physical activity target by 2030 [1,2,20].

In contrast, some countries such as Botswana (8.9% vs 7.7%), Ghana (10.9% vs 8.8%), Namibia (11.1% vs 10.9%), Sudan (6.2% vs 5.3%), Uganda (10.9% vs 8.9%), Zambia (7.5% vs 7.5%), and Zimbabwe (9.1% vs 8.6%) have similar or narrowed gender gap with respect to sufficient physical activity engagement. This result was purely exploratory and the reasons for the narrowed/similar gender gap in these countries are unknown to the authors. Therefore, further research is particularly important to explore the deterministic factors and reasons for the narrowing gap in these countries as it would provide helpful information in shaping intervention to bridge the gaps existing in all the countries included in the present study.

Heterogeneity between individual country surveys in the pooled prevalence of sufficient physical activity level was high, extending $I^2 >87\%$. As a result, sources of heterogeneity were further explored through sub-group and meta-regression analyses with respect to HDI, countries' economic health, sub-region, and survey years. The only possible source heterogeneity

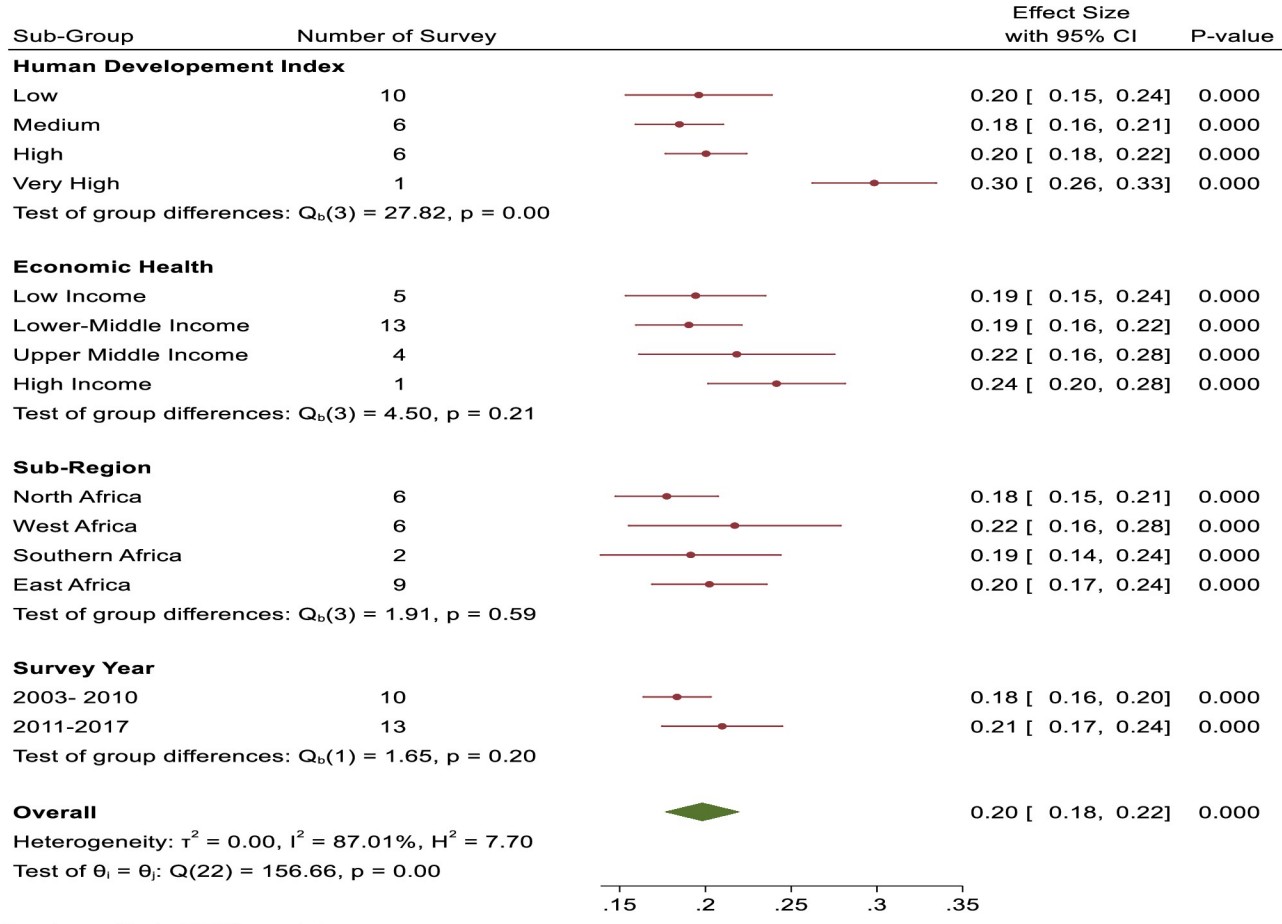

**Fig 4. Sub- group analysis for adolescent sufficient physical activity level with respect to HDI, economic health, sub-region and the survey years.**

was the survey years. For instance, in the sub-group analysis, surveys performed between 2003–2010 had sufficient physical activity level of 18% as compared to surveys conducted between 2011–2017 (i.e., 22%). Similarly, the multivariable meta-regression showed increased trends across the survey years.

## Strength and limitation

The current study explored comparatively large dataset from the GSHS which makes it generalizable to in-school adolescents in Africa. Meta-analysis with random effect was employed to pool the prevalence of physical activity level to cater for any heterogeneity that might have arisen from the data from the various countries. The physical activity variable was self-reported hence, there is a chance of reporting bias (i.e., could influence the under-estimation or over-estimation of the variables). Finally, the GSHS only included in-school students hence, results are not representative of the entire adolescent population in Africa.

## Conclusion

The current study based on over 64,000 in-school adolescents, is the first to estimate physical activity level across 23 African countries. Only 20% of in-school adolescents met the WHO's recommended physical activity level. Generally, adolescent girls in Africa are less active than

adolescent boys. Addressing the rising burden of insufficient physical activity in adolescents and narrowing the gender gap could ultimately increase the overall physical activity engagement and achieve the WHO's global physical activity target by 2030.

## Supporting information

**S1 Table. Leave one sensitivity analysis of the survey years.**
(DOCX)

**S2 Table. Meta-regression analysis to explore the sources of heterogeneity.**
(DOCX)

## Acknowledgments

This paper uses data from the Global School-Based Student Health Survey (GSHS). GSHS is supported by the World Health Organization and the US Centers for Disease Control and Prevention.

## Author Contributions

**Conceptualization:** Martin Ackah.

**Data curation:** Martin Ackah, David Owiredu.

**Formal analysis:** Martin Ackah.

**Investigation:** Martin Ackah, David Owiredu.

**Methodology:** Martin Ackah, David Owiredu, Mohammed Gazali Salifu, Cynthia Osei Yeboah.

**Project administration:** Martin Ackah.

**Resources:** Martin Ackah.

**Software:** Martin Ackah, David Owiredu.

**Supervision:** Martin Ackah.

**Validation:** Martin Ackah, David Owiredu, Cynthia Osei Yeboah.

**Visualization:** Martin Ackah, Mohammed Gazali Salifu.

**Writing – original draft:** Martin Ackah, David Owiredu, Mohammed Gazali Salifu, Cynthia Osei Yeboah.

**Writing – review & editing:** Martin Ackah, David Owiredu, Mohammed Gazali Salifu, Cynthia Osei Yeboah.

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
