## [Decision Letter · Decision Letter 0]

2 Aug 2022

PGPH-D-22-00968

Estimated prevalence and gender disparity of physical activity among 64,127 in-school adolescents (aged 12-17 years): A multi-country analysis of Global School-based Health Surveys from 23 African countries.

Dear Dr. Ackah,

Thank you for submitting your manuscript to PLOS Global Public Health. After careful consideration, we feel that it has merit but does not fully meet PLOS Global Public Health’s publication criteria as it currently stands. Therefore, we invite you to submit a revised version of the manuscript that addresses the points raised during the review process.

We look forward to receiving your revised manuscript.

Kind regards,

Martin Heine

Academic Editor

Journal Requirements:

Additional Editor Comments (if provided):

I agree with both reviewers that there is additional value in exploring heterogeneity and possibly more in-depth analysis of temporal or geographical trends within the data. Please don't hesitate to request more time for the revision if this is required to address the major feedback points adequately.

Reviewers' comments:

Reviewer's Responses to Questions

**Comments to the Author**

1. Does this manuscript meet PLOS Global Public Health’s publication criteria? Is the manuscript technically sound, and do the data support the conclusions? The manuscript must describe methodologically and ethically rigorous research with conclusions that are appropriately drawn based on the data presented.

Reviewer #1: Yes

Reviewer #2: Yes

2. Has the statistical analysis been performed appropriately and rigorously?

Reviewer #1: No

Reviewer #2: No

3. Have the authors made all data underlying the findings in their manuscript fully available (please refer to the Data Availability Statement at the start of the manuscript PDF file)?

Reviewer #1: No

Reviewer #2: No

4. Is the manuscript presented in an intelligible fashion and written in standard English?

Reviewer #1: Yes

Reviewer #2: Yes

5. Review Comments to the Author

Reviewer #1: This is an important analysis of multi country data on physical activity among adolescents in Africa.

Kindly find below my comments:

• This important analysis will miss a great opportunity to conduct subgroup analysis on the various sub-regions in Africa and looking at the trend of physical activity over the years.

• What was the age range of the adolescents? Was it similar across countries?

• Comment on the results of the heterogeneity test

• What are the possible sources of heterogeneity from the various countries? Discuss them and how they influence your results.

• Did you conduct any sensitivity analysis on the pooled prevalence? E.g. leave-one-out analysis? Such results will be very informative.

• Some of the data was conducted as far back as 2003. How relevant is such information to the pooled prevalence you estimated? How can the pooled prevalence inform decision making?

• Discuss on the impact of the differences in year of the data collection on the pooled prevalence

Minor comments

• Kindly check the use of abbreviations. Define them on first mention.

Reviewer #2: Introduction

1. Second sentence of Introduction, if words limit allow, could you give an example of the enormous benefits on health?

2. GSHS abbreviations in the manuscript are not quite consistent, I think you meant “Global School-based student Health Survey”?

3. Paragraph 3, last sentence, could you elaborate more on why the lack of representative African adolescents’ PA data is a problem?

Methods

4. I suggest the methods section should follow the STROBE guideline. If it is already following the guideline, please state.

5. I suggest you mention in the Measure subsection that the outcome variable, physical activity, was self-reported. You included this in the strength and limitation but not in the methods.

Results

6. Table 1, could you elaborate on the unit of physical activity level measurement? I am not sure if these are supposed to be minutes, or are they comparable with this format? Also add unit to response rate. Same comment for Table 2

7. Table 1: to make this more intuitive for the reader could the authors arrange with some logic eg alphabetical or in ascending order for any of the columns. Same comment for Table 2

8. Table 1, for sample size of each country, I would suggest that you also include the number of populations the sample size represented. Since in the statistical analysis section you mentioned: “The weights allow the results to be extrapolated to the entire study population as well as the entire in-school adolescents’ population in Africa”. So, the reader can get an idea of represented population.

9. Figures 1-3, I really like these figures however the image quality is poor. I wonder if this is due to technical issue.

10. Figures 2-3, from the methods section, I was not aware that there will be 3 meta-analysis figures. It would be helpful to elaborate how “burden of PA” was generated in both male and females in the methods section.

11. Figures 2-3, it would be helpful to elaborate what you mean by “burden of PA” in the methods section.

12. Figures 2-3, would it be possible to overlay these male and female figures so that it would be easier to compare them?

13. Given the evidence that inactivity increases with age, and variably by sex, could the authors comment on the age distribution of their sample? And whether there was any different in the age distrubtion in boys vs girls?

Discussion

14. I think the second sentence of the previous paragraph and the first sentence of this paragraph says the same thing?

15. I find this sentence “Physical inactivity among adolescents has become a public health epidemic around the world”, rather odd here. I think it would be nice for this sentence “Overall, only 20% of adolescents in Africa met the current WHO physical activity recommendation” to be followed directly by “This conclusion..”

16. 2nd paragraph, I wonder if you can make this paragraph more concise.

17. 2nd paragraph, I believe this sentence you be in Introduction instead.

18. 3rd paragraph, do you have any hypothesis why gender gap in Benin, Algeria, Tunisia and Mozambique are more pronounced than other African countries in this study?

19. 3rd paragraph, 6th sentence, this sentence does not quite support the whole paragraph whole argument since it is talking about behaviour modification. I would suggest focusing on what could drive the gender gap in these countries. Also, it would be helpful to show and discuss countries which are more equal or even have female adolescents engaging in more physical activity compared to the male counterparts.

20. 3rd paragraph, last sentence, while this sentence is more supportive of the whole paragraph, but the conclusion on social, cultural, and so on is disconnected from the study.

21. There is no discussion on heterogeneity? Could the authors consider any other factors that may bias findings? For eg were all included schools in urban areas? And if not how do these findings differ between urban and rural? And across different socio-economic levels? This is the most significant weakness of the paper and it makes it challenging to rigorously interpret the findings

Conclusion

22. Last sentence, same concern the recommendation is disconnected from the study.

6. PLOS authors have the option to publish the peer review history of their article (what does this mean?). If published, this will include your full peer review and any attached files.

**Do you want your identity to be public for this peer review?** For information about this choice, including consent withdrawal, please see our Privacy Policy.

Reviewer #1: No

Reviewer #2: No

---

## [Decision Letter · Decision Letter 1]

27 Sep 2022

Estimated prevalence and gender disparity of physical activity among 64,127 in-school adolescents (aged 12-17 years): A multi-country analysis of Global School-based Health Surveys from 23 African countries.

PGPH-D-22-00968R1

Dear Mr. Ackah,

We are pleased to inform you that your manuscript 'Estimated prevalence and gender disparity of physical activity among 64,127 in-school adolescents (aged 12-17 years): A multi-country analysis of Global School-based Health Surveys from 23 African countries.' has been provisionally accepted for publication in PLOS Global Public Health.

Best regards,

Martin Heine

Academic Editor

Reviewer Comments (if any, and for reference):

Reviewer's Responses to Questions

**Comments to the Author**

1. If the authors have adequately addressed your comments raised in a previous round of review and you feel that this manuscript is now acceptable for publication, you may indicate that here to bypass the “Comments to the Author” section, enter your conflict of interest statement in the “Confidential to Editor” section, and submit your "Accept" recommendation.

Reviewer #1: All comments have been addressed

2. Does this manuscript meet PLOS Global Public Health’s publication criteria? Is the manuscript technically sound, and do the data support the conclusions? The manuscript must describe methodologically and ethically rigorous research with conclusions that are appropriately drawn based on the data presented.

Reviewer #1: Yes

3. Has the statistical analysis been performed appropriately and rigorously?

Reviewer #1: Yes

4. Have the authors made all data underlying the findings in their manuscript fully available (please refer to the Data Availability Statement at the start of the manuscript PDF file)?

Reviewer #1: Yes

5. Is the manuscript presented in an intelligible fashion and written in standard English?

Reviewer #1: Yes

6. Review Comments to the Author

Reviewer #1: My comments have been addressed.

7. PLOS authors have the option to publish the peer review history of their article (what does this mean?). If published, this will include your full peer review and any attached files.

**Do you want your identity to be public for this peer review?** For information about this choice, including consent withdrawal, please see our Privacy Policy.

Reviewer #1: **Yes: **Daniel Boateng
